# Aluminum Nitride Ultraviolet Light-Emitting Device Excited via Carbon Nanotube Field-Emission Electron Beam

**DOI:** 10.3390/nano13061067

**Published:** 2023-03-16

**Authors:** Yangcheng Yu, Dong Han, Haiyuan Wei, Ziying Tang, Lei Luo, Tianzeng Hong, Yan Shen, Huying Zheng, Yaqi Wang, Runchen Wang, Hai Zhu, Shaozhi Deng

**Affiliations:** 1State Key Laboratory of Optoelectronic Materials and Technologies, School of Physics, Sun Yat-sen University, Guangzhou 510275, China; 2State Key Laboratory of Optoelectronic Materials and Technologies, Guangdong Province Key Laboratory of Display Material and Technology, School of Electronics and Information Technology, Sun Yat-sen University, Guangzhou 510275, China

**Keywords:** AlN, carbon nanotube, field emission, ultraviolet light-emitting device

## Abstract

With the progress of wide bandgap semiconductors, compact solid-state light-emitting devices for the ultraviolet wavelength region are of considerable technological interest as alternatives to conventional ultraviolet lamps in recent years. Here, the potential of aluminum nitride (AlN) as an ultraviolet luminescent material was studied. An ultraviolet light-emitting device, equipped with a carbon nanotube (CNT) array as the field-emission excitation source and AlN thin film as cathodoluminescent material, was fabricated. In operation, square high-voltage pulses with a 100 Hz repetition frequency and a 10% duty ratio were applied to the anode. The output spectra reveal a dominant ultraviolet emission at 330 nm with a short-wavelength shoulder at 285 nm, which increases with the anode driving voltage. This work has explored the potential of AlN thin film as a cathodoluminescent material and provides a platform for investigating other ultrawide bandgap (UWBG) semiconductors. Furthermore, while using AlN thin film and a carbon nanotube array as electrodes, this ultraviolet cathodoluminescent device can be more compact and versatile than conventional lamps. It is anticipated to be useful in a variety of applications such as photochemistry, biotechnology and optoelectronics devices.

## 1. Introduction

Ultraviolet light-emitting devices have attracted intensive scientific attention due to their wide range of applications in solid-state lighting, water sterilization, biomedical analysis, microelectronic photolithography and high-density data storage [1,2,3,4,5,6,7,8,9]. Nevertheless, the most frequently utilized ultraviolet light sources, excimer and mercury lamps, have the disadvantages of restricted portability and the presence of harmful constituents. Portable and environmentally-friendly alternative ultraviolet light-emitting devices are therefore being sought out. Solid-state ultrawide bandgap (UWBG) semiconductors, such as hexagonal boron nitride (hBN) [10] and III–V nitride semiconductors (aluminum gallium nitride (AlGaN), aluminum nitride (AlN)) [11,12], show promise as materials for this purpose. For UWBG materials, trying to manufacture light-emitting diodes (LEDs) is a challenging task due to doping (especially for p-type doping) and ohmic contact issues [13]. In this context, electron beam (EB) pumping is a fascinating strategy, and is compatible with UWBG semiconductors. In an EB-pumped light-emitting device, the operating principle is different from that of LEDs, and a p-n junction is not required. After applying a high voltage, electrons escape from the cathode and are accelerated to bombard the anode, which is also the cathodoluminescent material. Electron-hole pairs are generated in the anode semiconductor by the impact-ionization processes of the incoming accelerated electrons. Then, these excited electrons relax to a lower state via radiative recombination and result in photon emission [14]. Compared with LEDs, it is obvious that there is no need to take doping and contacts into account in an EB-pumped device.

Al_x_Ga_1-x_N/AlN multiple quantum wells (MQWs) material system is the main research topic on EB-pumped light sources, with an emission wavelength varying from 210 to 365 nm by adjusting the Al molar fraction x [15,16,17,18,19,20]. Oto et al. first reported 240 nm deep-ultraviolet emission on Al_0.69_Ga_0.31_N/AlN MQWs pumped by EB [15], and since then a series of efforts on AlGaN/AlN MQWs have been reported successively, with different wavelengths obtained. In addition, taking advantage of the highly luminous properties of hBN, a handheld ultraviolet device equipped with a field-emission array, was demonstrated on hBN powder with an excitonic emission at 225 nm [21]. Furthermore, a cathodoluminescent device using exfoliated hBN as a target for EB was fabricated with a defect-related emission at 350 nm [10]. To the best of our knowledge, no demonstration of an EB-pumped ultraviolet light-emitting device based on AlN thin film has been reported. In this work, we designed and demonstrated an AlN-based ultraviolet cathodoluminescent device. The cold cathode is constructed by a carbon nanotube (CNT) array, which is favorable for achieving high electron emission efficiency through the field emission process and possesses many advantages over traditional hot cathodes [17]. The output spectra showed a dominant ultraviolet peak at 330 nm with a short-wavelength shoulder at 285 nm and were analyzed in detail. The light output intensity of our device did not display saturation under the present driving voltage, indicating that the output intensity still has a large promotion space. Our results validated the feasibility of EB pumping for the AlN-based ultraviolet light-emitting device and provide a platform for investigating other UWBG semiconductor materials.

## 2. Materials and Methods

The CNTs were fabricated using a thermal chemical vapor deposition (TCVD) growth method [22,23,24]. A mask with hole area of 1 × 1 mm^2^ was attached to a cleaned silicon substrate, and 12 nm alumina and 1.5 nm iron were sputtered by Gatan ion coater as catalysts. Then, the silicon substrate was put in the TCVD furnace. While the CNTs were growing, a mixed gas source of C_2_H_4_, H_2_ and Ar was introduced onto the substrate at 750 °C for more than 30 min in the furnace. The flow rates of C_2_H_4_, H_2_ and Ar were set at 50 sccm, 100 sccm and 400 sccm, respectively. The morphologies of CNTs were investigated by scanning electron micrography (SEM, Hitachi S-4800; Hitachi Ltd., Tokyo, Japan) and transmission electron micrography (TEM, FEI Tecnai G2 F30; FEI Inc., Washington, DC, USA), and the Raman spectrum was obtained at room temperature (RT) using a He–Ne (633 nm) laser (LabRAM HR; Horiba Ltd., Kyoto, Japan).

The AlN thin film was grown on a 2-inch c-plane sapphire substrate by metal organic chemical vapor deposition, with trimethylaluminum and ammonia as precursors. To facilitate further use, the AlN sample was cut into pieces with an area of 5 × 5 mm^2^ using a high-power Nd:YAG (266 nm) laser. The morphologies of the AlN sample were characterized by SEM and atomic force micrography (AFM, nanoIR2-s; Anasys Instruments Inc., Santa Barbara, CA, USA). The high-resolution TEM (HRTEM, FEI Tecnai G2 F30; FEI Inc., Washington, DC, USA) image, selected area electron diffraction (SAED, FEI Tecnai G2 F30; FEI Inc., Washington, DC, USA) pattern and x-ray diffraction (XRD, Supernova; Rigaku Co., Tokyo, Japan) pattern were performed to investigate crystal quality. To function normally, the homemade ultraviolet light-emitting setup was placed in a vacuum chamber with a pressure of 1 × 10^−6^, and variable square high-voltage pulses lower than 4 kV with a 100 Hz repetition frequency and a 10% duty ratio were applied to the anode. A sensitive spectrometer (QE Pro; Ocean Optics Inc., Dunedin, FL, USA) with an ultraviolet optical fiber was used to collect and detect the output spectra.

## 3. Results and Discussion

### 3.1. CNT Emitter

The CNTs are used as the electron emitter media due to their advantages in low turn-on electric fields, highly stable emission current, and environmentally-friendly nature. A typical SEM image presents the as-synthesized vertically aligned CNT cube, which displays the uniform top surface and side walls with a height exceeding 600 μm (Figure 1a). To further explore the morphology of CNTs, the top ends and stems of CNTs are shown at higher magnifications in the SEM photos (Figure 1b,c). It can be seen that these prepared CNTs show sharp tips and are arranged vertically. In addition, as shown in the TEM image (Figure 1d), the CNTs are observed to have few carbon layers (typically less than five layers) and an average diameter estimated to be roughly 10 nm. Comprehensive analyses of the SEM and TEM investigations indicate that the CNTs possess a high aspect ratio and strong localized field enhancement effect, making them appropriate for field electron emission, particularly in a low turn-on electric field and large current (high current density) situations.

The Raman spectrum of the synthesized CNTs is displayed in Figure 1e to evaluate the crystalline perfection. The Raman peak at 1592 cm^−1^ is introduced by the C−C stretching mode and is known as G peak. Another peak at 1339 cm^−1^, commonly referred to as the D peak, can be ascribed to the defects and amorphous carbon existing in the CNT emitter [25]. The intensity ratio of the G peak and D peak (I_G_/I_D_) was measured to be around 1. The I_G_/I_D_ is frequently used to access the crystalline quality of CNTs. The CNTs with a large I_G_/I_D_ usually possess high crystalline quality and display efficient field emission [26]. No signal of single-wall CNT radial breathing mode was observed, indicating that the prepared CNTs have a multi-layer structure, which is in accordance with TEM characterization. The measured field emission current density versus electric field curve of the CNT emitter is depicted in Figure 1f. It shows a low turn-on electric field of about 3.5 V/μm.

### 3.2. AlN Film

Figure 2a gives the SEM image of the AlN sample cross-section. The AlN thin film was grown on a sapphire substrate and the thickness of the AlN layer is about 380 nm. To assess the surface roughness of the AlN layer, a three-dimensional AFM picture (Figure 2b) with a scan size of 5 × 5 μm^2^ was taken. The AlN layer shows uniform, smooth and dense surface topography, with a root-mean-square roughness value of 0.4 nm. Furthermore, TEM and XRD have been widely utilized to characterize crystal quality. The cross-sectional HRTEM image of the AlN film sample and the corresponding SAED pattern of the AlN layer are shown in Figure 2c. The SAED pattern is a spot pattern, which reveals that the AlN film is a single crystal in nature with a wurtzite hexagonal phase [27]. According to the HRTEM, the d-spacing is measured to be 0.496 nm in the AlN layer and 0.430 nm in the sapphire substrate, which matches the typical d-spacing values of AlN (001) and sapphire (003), respectively [28,29]. This means the (001) planes of the hexagonal AlN phase are parallel to the substrate and thus to the film surface. In other words, the AlN film was grown along the *c*-axis on a c-plane sapphire substrate.

Figure 2d displays the XRD pattern of the AlN sample, in which two strong diffraction peaks can be discovered. The peak at 2θ = 36.05° corresponds to the (002) reflex of wurtzite structure [30,31], suggesting the AlN layer is grown along the *c*-axis. Furthermore, the substrate is a c-plane sapphire, as seen by the peak at 2θ = 41.75°, which is associated with the (006) plane [31]. This result is consistent with the TEM analysis. It should be noted that the (002) diffraction peak in the XRD pattern of hexagonal wurtzite AlN corresponds to the second-order diffraction of the (001) plane. The absence of odd (00*l*) diffraction peaks, such as (001) and (003) peaks, is due to the extinction rule. For a hexagonal lattice, the extinction rule can be expressed in the following expression:*h* + *2k* = *3n*(1)
where (*hkl*) is the diffraction index for an XRD diffraction peak, *l* is an odd number, and *n* is an integer. The peaks that satisfy expression (1) will not appear in the XRD pattern. Thus, the (001) diffraction peak cannot be observed in the XRD pattern while the typical d-spacing value of AlN (001) is measured in the HRTEM image. A similar analysis also can be applied to the sapphire substrate based on its lattice structure and extinction rule. The inset of Figure 2d shows the rocking curve of the AlN (002) plane with a full width at half maximum (FWHM) of 144 arcsec, which is in the range of typical values (90 arcsec to 400 arcsec) [32] and indicates good crystalline quality in the AlN thin film. 

The Tauc method [33] was used to determine the bandgap of the AlN sample, which is based on the absorption coefficient α and can be described by the following equation:
(2)$$\left(\alpha hv\right){}^{n}=B\left(hv-E_{g}\right)$$
where *B* is a constant, *ν* is the frequency of the photon, *h* is the Plank constant, *E_g_* is the bandgap, *n* = 1/2 for indirect bandgap semiconductor and *n* = 2 for direct bandgap semiconductor. As the AlN is a direct bandgap semiconductor, we take *n* = 2. As shown in Figure 3, an absorption spectrum of the AlN sample has been obtained, and the inset exhibits the corresponding Tauc plot with a linear fit extrapolated to the *x*-axis. According to the above equation, the *x*-axis intersection point of the linear fit gives an estimate of the bandgap value of the AlN sample, which is measured to be 6.11 eV.

### 3.3. Prototype Device

Figure 4 displays the schematic diagram and photograph of our homemade cold-cathode EB-pumping ultraviolet light-emitting device. The cathodoluminescent material, namely AlN thin film with an area of 5 × 5 mm^2^, is directly mounted onto the center of the anode component while the electron emitters, namely CNT cubes, are transferred and evenly distributed on the cathode component at an inward slope (Figure 4b), fixed with conductive silver paste. Here, a cathode-anode diode driving structure with a very short vacuum gap (600 μm) is adopted in this device to ensure the collimation of the incident EB and the energy density. Simultaneously, this compact core component suggests that the volume of the whole device can be further reduced with a simplified construction. The pictures for individual components and the assembly process of the ultraviolet device can be seen in Appendix A.

In operation, the entire device is insulated from its surroundings by a glass tube and loaded into a vacuum chamber with a sapphire exit window. The operating circuit in this work is shown in Appendix A. After applying a driving voltage of no more than 4 kV, the EB generated from the cold cathode bombards and excites the AlN thin film. An oscilloscope with a resistance was used to display the driving voltage waveform and calculate the current. The ultraviolet light generated via radiation recombination is designed to emit from the side of the sample surface through a central aperture of the cathode. In this way, transparency of the electrode and substrate is not necessary. Then, the light beam divergence is improved by an ultraviolet aspherical lens, to facilitate the subsequent collection and detection of the light signal.

### 3.4. Spectra Analysis

Figure 5a shows the cathodoluminescence (CL) spectra of the ultraviolet device under variable driving voltages. An ultraviolet emission with a short-wavelength shoulder dominates the spectra. As illustrated in a semi-logarithmic plot (inset of Figure 5a), the ultraviolet band of the output spectra contains two emission peaks: one weak deep ultraviolet peak centered at 285 nm and one strong ultraviolet peak centered at 330 nm. According to earlier photoluminescence and electroluminescence studies, the emission at 285 nm (4.35 eV) can be attributed to the transition from the conduction band to the aluminum vacancy complex [9,34], while the emission at 330 nm (3.75 eV) has been demonstrated to be related to the transition between the shallow donor and aluminum vacancy complex [35,36,37]. More specifically, the transition behavior of electrons can be described as follows: when the AlN emitting layer is excited, most of the excited electrons are captured by the shallow donor and then recombine with the holes on the energy levels related to the aluminum vacancy complex, resulting in a strong emission at 330 nm (3.75 eV). A small number of electrons are excited to the conduction band and contribute to the weak emission at 285 nm (4.35 eV) by recombining with holes on energy levels related to the aluminum vacancy complex. No conduction band electron recombines with holes in the valence band, so the near bandgap emission of AlN is absent in the CL spectra.

A photo of the light emission is necessary because it can enable the readers to gain an understanding of the device’s luminous effect. As shown in Figure 5b, a photo of the device was taken using an ultraviolet-sensitive camera when the device was in operation. In this photo, the outer circle was caused by the light scattering from the edge of the ultraviolet lens used to improve light beam divergence (shown in Figure 4a). The luminescent spot of the device was located inside the circle, with a shape of cross which was caused by four symmetrically distributed CNT cubes (shown in Figure 4b). To estimate the power efficiency (PE) of the ultraviolet device, the light output power needs to be measured. A schematic diagram of the light output power measurement setup is shown in Appendix A. The light emitted from the AlN thin film first passed through a 1-inch ultraviolet lens to improve beam divergence and then exited from the vacuum chamber through a 2-inch sapphire window. A 2-inch ultraviolet lens was placed close to the sapphire window to focus the light beam. Finally, the output power was measured by a power meter. When a driving voltage with an amplitude of 3.92 kV and a duty ratio of 10% (namely, V_d_ = 3.92 kV and δ = 10%) was applied, the corresponding current was 1.6 mA (I_d_ = 1.6 mA) with the same duty ratio. Thus, the input electrical power (P_in_) was about 62 mW, which can be calculated using the expression P_in_ = I_d_ V_d_ δ^2^. Under this driving voltage, the light output power (P_out_) was measured to be about 10 μW. Consequently, the power efficiency (PE) of the device, calculated by the equation PE = P_in_/P_out_, was estimated to be about 0.02%.

The evolution of light output intensity with increasing excitation voltage is shown in Figure 5c. It reveals that the light output intensity increases with the excitation voltage, which can be explained as follows: first, as displayed in Figure 1f, when the electric field exceeds the turn-on electric field, the current density increases rapidly. This means that a higher driving voltage in the device will enable more electrons to escape from the cathode and participate in the CL process, resulting in stronger light output intensity. A further CL experiment with different duty ratios (δ) was performed to show the impact of the quantity of the electrons on the output intensity. The CL spectra for δ = 10% and 20% are shown in Appendix A. For ease of comparison, the CL spectra, excited by driving voltage with the same amplitude but different duty ratios, are marked with the same color. It can be observed that the emission peaks are stronger with a larger duty ratio. Under a driving voltage with the same amplitude, a larger duty ratio allows more electrons to escape from CNTs and participate in the CL process in one cycle time. In an ideal condition, the output intensity with a duty ratio of 20% is twice that with a duty ratio of 10%. However, the accumulation of negative charge caused by the poor conductivity of the AlN thin film may weaken the electron emission, which is more evident with a larger duty ratio. Taking this charge shielding effect and measurement error into consideration, the actual result (as shown in Appendix A) has a little deviation from the ideal condition.

On the other hand, when electrons are accelerated from the cathode to the anode, they will get higher energy under a larger driving voltage, allowing them to penetrate deeper into the sample and generate more electron-hole pairs through impact-ionization processes. To be more specific, Figure 6 presents the simulations of electron trajectory injected in the AlN thin film sample under different driving voltages (3, 3.5 and 4 kV) using Monte Carlo method (software ‘CASINO’, v2.5.1.0, Sherbrooke, Canada) [38]. In the simulation, the thickness of the AlN layer was set as 380 nm, which was consistent with the actual value of the AlN sample measured by SEM. The densities of AlN and sapphire were set as 3.23 g/cm^3^ and 4.00 g/cm^3^, respectively. The electron energy loss, roughly representing CL emission intensity, is depicted in the inset as a function of the penetration depth. It can be observed that increasing the driving voltage leads to a larger penetration depth. Under the largest driving voltage (4 kV), the penetration depth is less than the thickness of the AlN layer, which means almost all electrons contribute to CL emission, instead of taking an energy loss in the substrate. Finally, it can be seen that the output intensity is the result of the combined effect of electron energy and quantity. Moreover, the light output intensity in Figure 5c did not exhibit a saturation phenomenon under the present driving voltage, indicating that the output intensity has a large promotion space. With a higher voltage controlled within a reasonable range, which can enable more electrons to escape from the cathode and participate in the CL process with higher energy, a higher intensity of ultraviolet emission is expected.

## 4. Conclusions

In this study, we have explored the potential of AlN thin film as cathodoluminescent material. An ultraviolet light-emitting prototype device with the AlN thin film as a target for EB pumping was designed and demonstrated. The fabricated CNTs were employed as field-emission media benefitting from their low turn-on electric field. The CL spectra, increasing with the driving voltage, showed a dominant ultraviolet emission at 330 nm with a short-wavelength shoulder at 285 nm and were discussed in detail. The ultraviolet device achieved an output power of 10 μW and power efficiency of 0.02 % under a driving voltage of 3.92 kV. In addition, the light output intensity did not exhibit a saturation phenomenon with increasing driving voltage, indicating that the output intensity has a large promotion space. This work validates the EB pumping strategy for luminescence of AlN thin film and provides a platform for investigating other UWBG semiconductors. Furthermore, the cathode-anode driving structure with a short gap suggests that the volume of the device can be further reduced with a simplified construction. The device can be more compact and versatile than conventional lamps and is anticipated to be useful in a variety of ultraviolet applications.

## Figures and Tables

**Figure 1 nanomaterials-13-01067-f001:**
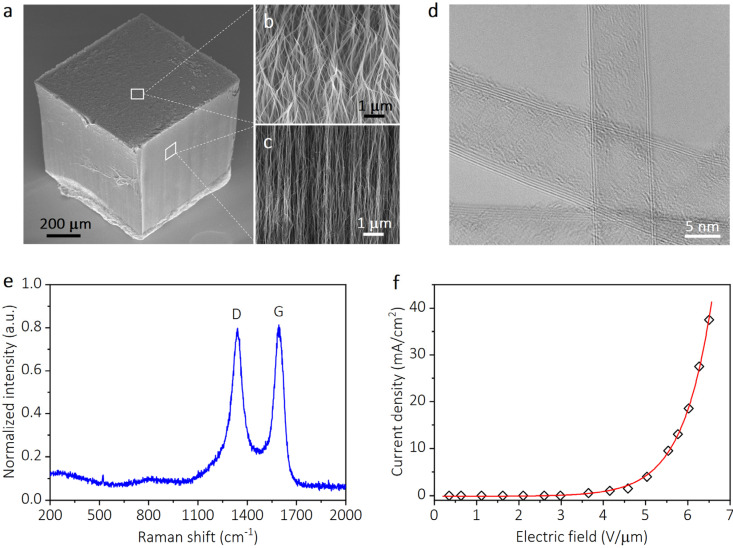
The characteristics of the carbon nanotube (CNT) emitter. (**a**) A typical scanning electron micrography (SEM) image of the as-synthesized cube-shaped CNT emitter. (**b**,**c**) High-magnification SEM micrographs of CNTs with areas marked in (**a**), exhibiting the sharp top ends and the vertically aligned stems of CNTs. (**d**) Transmission electron micrography (TEM) image of CNTs with few walls. (**e**) Raman spectrum of CNT emitter with D and G peaks labeled. (**f**) Field emission current density versus applied electrical field measured from CNT emitter.

**Figure 2 nanomaterials-13-01067-f002:**
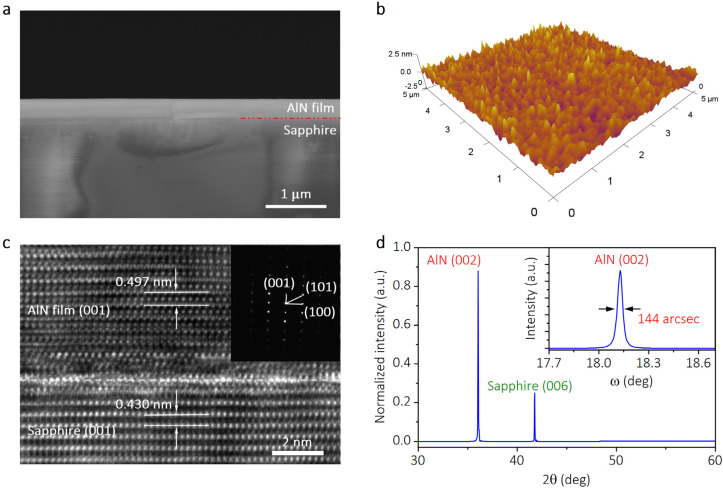
The characterizations of aluminum nitride (AlN) thin film. (**a**) Cross-sectional SEM image of the AlN thin film grown on sapphire substrate. (**b**) AFM image of the AlN thin film, with a scan size of 5 × 5 μm^2^. (**c**) Cross-sectional HRTEM image of the AlN film sample, with the corresponding selected area electron diffraction (SAED) pattern of the AlN layer shown in the inset. (**d**) X-ray diffraction (XRD) pattern of the AlN sample showing two diffraction peaks, corresponding to AlN (002) and sapphire (006). The inset shows the rocking curve of the AlN (002) plane.

**Figure 3 nanomaterials-13-01067-f003:**
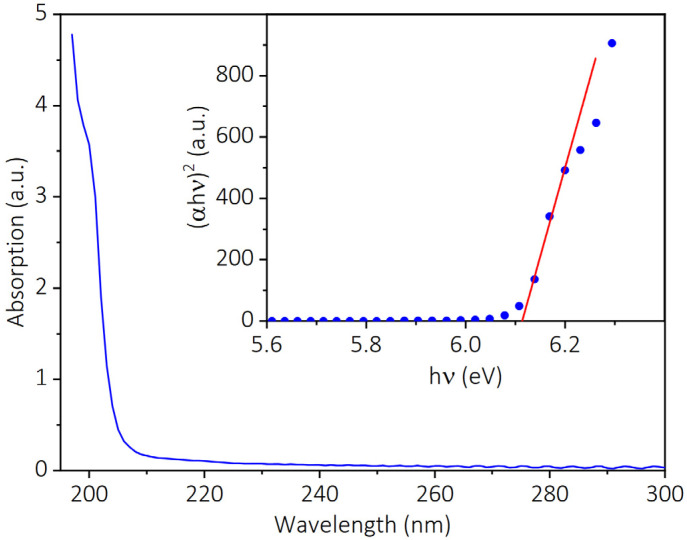
The absorption spectrum of the AlN sample. The inset is the corresponding Tauc plot.

**Figure 4 nanomaterials-13-01067-f004:**
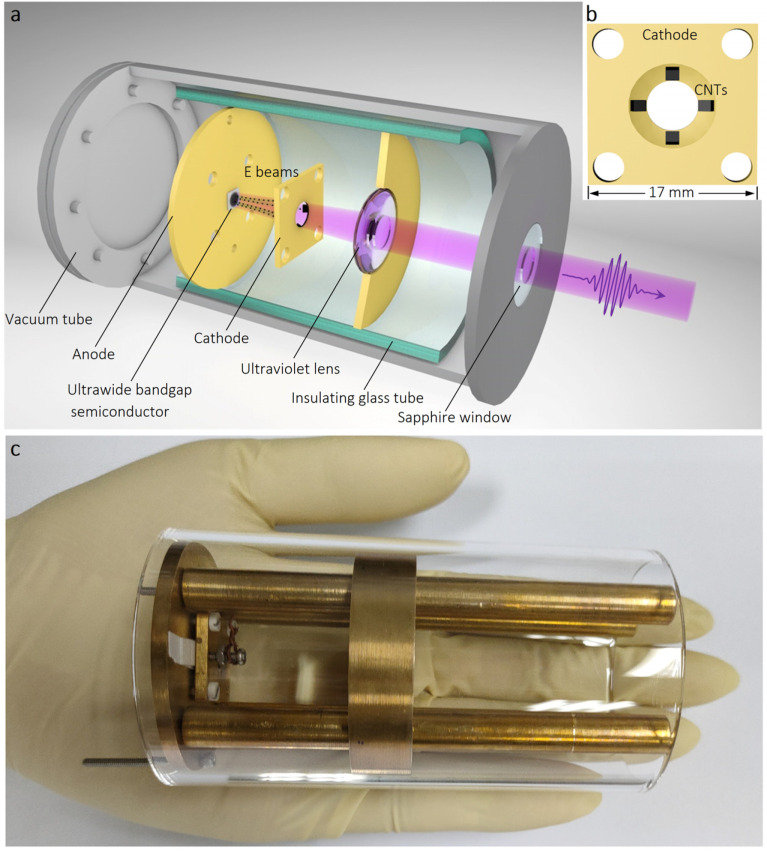
Ultraviolet cathodoluminescent device configuration. (**a**) Schematic diagram of the EB-pumping ultraviolet light emission setup. (**b**) Diagrammatic representation of cold cathode structure with four CNT cubes attached symmetrically. (**c**) A digital image of the ultraviolet cathodoluminescent device protype.

**Figure 5 nanomaterials-13-01067-f005:**
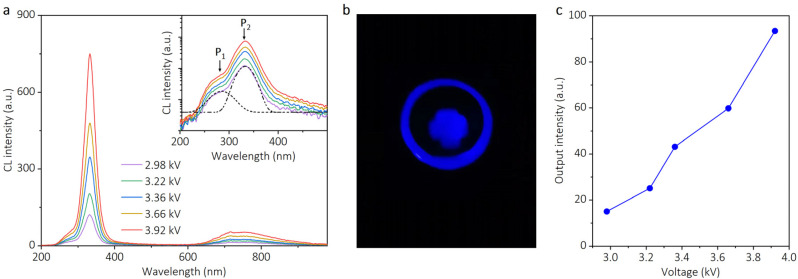
Luminescence properties of the ultraviolet device. (**a**) The output spectra of the device at various excitation voltages. The inset displays a semi-logarithmic plot of the output spectra in the short-wavelength region, with two Gaussian fit peaks marked as P_1_ and P_2_. (**b**) A photograph of the device taken by an ultraviolet-sensitive camera while it is in operation. (**c**) The variation of output intensity with excitation voltage.

**Figure 6 nanomaterials-13-01067-f006:**
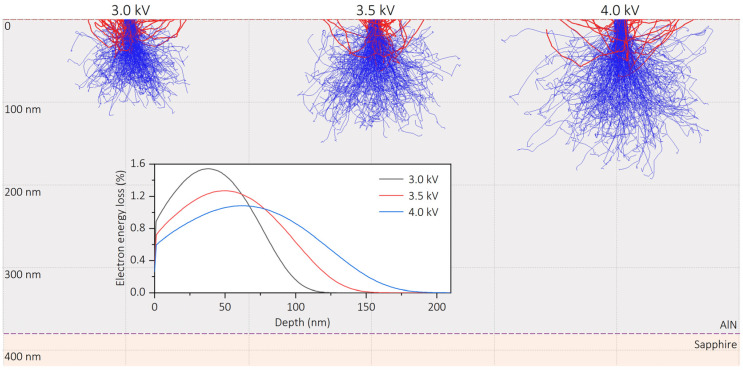
Monte Carlo Simulations of electron trajectory in the AlN sample under different excitation voltages, in which the blue trajectories originate from absorbed electrons and the red trajectories come from backscattered electrons. The inset represents the electron energy loss as a function of the penetration depth in the AlN sample.

## Data Availability

The data presented in this study are available on request from the corresponding author.

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
