# Peer review of "Aluminum Nitride Ultraviolet Light-Emitting Device Excited via Carbon Nanotube Field-Emission Electron Beam"

_nanomaterials, 2023, doi:10.3390/nano13061067_

Round 1
Reviewer 1 Report
Comments
This work shows the UV-light-emitting device using a TCVD AlN thin film as the cathodoluminescence material. The well-set-up and clear Figures were given with consistent explanations.
However, there are still some confusing problems in this work.
1. Initially, the D peak (defects or amorphous indicator) in Raman is approximately equal to the stretching C-C mode (G peak), as seen in Figure 1e. This means that the grown multi-layer CNTs contained a high defect concentration or were not good crystalline material. The ref. 22 that you used for the growth of CNTs indicated a D/G peak ratio in Raman of roughly 0.73, which is quite low compared to your Raman results.
The XRD peaks of AlN are consistent with the previous report, where 2 theta is 36.050 as the (002) reflex of the wurtzite structure. In addition, the TEM result showed a d-spacing of 0.496 nm, which matches the typical values of AlN (001).
2. I guess due to the high defect concentration in CNTs, as shown in Raman results, resulting in the low emission current densities (Figure 1f). Although there is a low onset electric field in your case (3.5 V/μm), however, the current density is a magnitude lower than the other work (for example, at the same 6 V/μm, the current density in this work is around 20 mA/cm2 versus 200 mA/cm2 in the reference paper).
Do you have any ideas for this phenomenon, and can you improve your CNTs-based emission current densities with the electric fields?
3. The various voltages (until 3.92 kV) have been applied that could increase the penetration depth of the electron beam (EB) into the light-emitting material (AlN) layer to enhance the UV-light emission intensity.
In such a case, how much influence of the driving voltage could affect the electron-beam-source materials (i.e. CNTs) and the emitting layer (AlN)?
4. In other words, how long can the device in this work maintain the light emission intensity with the same driving voltages, that is, the lifetime of the device?
For the practical application, it is better to show the CNT-based electron emitted source lifetime and AlN layer one to demonstrate the stability of the electron beam source and the emission material.
5. This paper explains that the emission at 285 nm (4.35 eV) originates from the transition from the conduction band to the aluminum vacancy and vacancy-impurity complex.
While the emission at 330 nm (3.75 eV) is attributed to the transition between the shallow donor and aluminum vacancy complex.
What is the bandgap in your AlN material?
6. I wonder whether we could increase the PL intensity of the device in this work by increasing aluminum vacancy concentration or not. Would you answer this?
7. PL peaks in AlN related to aluminum vacancy and vacancy-impurity complex defects. On the other hand, the previous work indicated the oxygen-defect-assisted PL peaks at 2.53 eV. (J. Appl. Phys. 2002, 41, L28).
Are the PL or EL peaks dependent on the defect types, that is, could we control these PL peaks with defect types?
8. With 4 kV driving voltage, the penetration depth is simulated by the Monte Carlo method around 200 nm.
Was the sample quality influenced by the voltage of 4 kV?
If we use the higher voltage, do you expect to see a higher intensity of UV emission?
Author Response
Dear reviewer,
Thank you for the enlightening suggestions and recommendations. The following replies have thoroughly addressed all of the comments in the revised manuscript.
Comments and Suggestions for Authors
This work shows the UV-light-emitting device using a TCVD AlN thin film as the cathodoluminescence material. The well-set-up and clear Figures were given with consistent explanations.
However, there are still some confusing problems in this work.
- Initially, the D peak (defects or amorphous indicator) in Raman is approximately equal to the stretching C-C mode (G peak), as seen in Figure 1e. This means that the grown multi-layer CNTs contained a high defect concentration or were not good crystalline material. The ref. 22 that you used for the growth of CNTs indicated a D/G peak ratio in Raman of roughly 0.73, which is quite low compared to your Raman results.
The XRD peaks of AlN are consistent with the previous report, where 2 theta is 36.050 as the (002) reflex of the wurtzite structure. In addition, the TEM result showed a d-spacing of 0.496 nm, which matches the typical values of AlN (001).
Response: In ref. 22, the CNTs were grown on stainless steel substrate with a 2 nm nickel film as catalyst, while the vertically aligned CNTs in this work were grown on silicon substrate with 12 nm alumina and 1.5 nm iron as catalysts. The D/G peak ratio in our Raman spectrum is higher than the one in ref. 22. It can be caused by different details in growth process, such as growth temperature, substrate and catalyst, and gas flow rate. To improve the quality of CNTs, we will make further efforts.
The (002) diffraction peak in the XRD pattern of hexagonal wurtzite AlN corresponds to the second-order diffraction of (001) plane. For a hexagonal lattice, the extinction rule is:
h + 2k = 3n, while l is an odd number
where (hkl) is the diffraction index for an XRD diffraction peak, and n is an integer. The (001) peak that satisfies the extinction rule. Thus, the (001) diffraction peak can not be observed in the XRD pattern while the typical d-spacing value of AlN (001) is measured in the HRTEM image.
The sentences “It should be noted that the (002) diffraction peak in the XRD pattern of hexagonal wurtzite AlN corresponds to the second-order diffraction of the (001) plane. The absence of odd (00l) diffraction peaks, such as (001) and (003) peaks, is due to the extinction rule. For a hexagonal lattice, the extinction rule can be expressed in the following expression:
h + 2k = 3n, while l is an odd number (1)
where (hkl) is the diffraction index for an XRD diffraction peak, and n is an integer. The peaks that satisfy expression (1) will not appear in the XRD pattern. Thus, the (001) diffraction peak can not be observed in the XRD pattern while the typical d-spacing value of AlN (001) is measured in the HRTEM image. A similar analysis also can be applied to the sapphire substrate based on its lattice structure and extinction rule.” have been added at line-144 in revised manuscript.
- I guess due to the high defect concentration in CNTs, as shown in Raman results, resulting in the low emission current densities (Figure 1f). Although there is a low onset electric field in your case (3.5 V/μm), however, the current density is a magnitude lower than the other work (for example, at the same 6 V/μm, the current density in this work is around 20 mA/cm2 versus 200 mA/cm2 in the reference paper).
Do you have any ideas for this phenomenon, and can you improve your CNTs-based emission current densities with the electric fields?
Response: The reviewer’s perspective is enlightening for us. The high D/G peak ratio indicates more defects existing in CNTs, which may affect the transmission and emission of electrons. Sveningsson et al. [R1] have mentioned in his work that a large IG/ID indicates high crystalline quality and the field emission is correspondingly efficient.
The most essential way to increase the current densities is to improve the quality of CNTs by finding a better growth process, which requires a lot of experiments and much time. A temporary method suitable for our device is to properly reduce the distance between the cathode and the anode so as to obtain a larger electric field and thus a larger current density at the same voltage. It is worth noting that the distance between the cathode and the anode should be carefully controlled. First, a too small distance may increase the possibility that some CNTs are pulled up from the cathode by the strong electric field and connect to the anode, resulting in a short circuit in the device. On the other hand, a strong electric field may lead to the field evaporation of CNTs, resulting in carbon deposition on the surface of AlN and affecting the luminescence of the device. For actual device application, it is necessary to take various influencing factors into consideration. In general, as the reviewer reminded, improving the quality of CNTs is the best choice for further research.
The sentences “The intensity ratio of the G peak and D peak (IG/ID) was measured to be around 1. The IG/ID is frequently used to access the crystalline quality of CNTs. The CNTs with a large IG/ID usually possess high crystalline quality and display efficient field emission [26].” have been added at line-118 in revised manuscript.
- The various voltages (until 3.92 kV) have been applied that could increase the penetration depth of the electron beam (EB) into the light-emitting material (AlN) layer to enhance the UV-light emission intensity.
In such a case, how much influence of the driving voltage could affect the electron-beam-source materials (i.e. CNTs) and the emitting layer (AlN)?
Response: For the electron-beam-source materials (i.e. CNTs), driving voltage provides an electric field for the electron beam to escape from the CNTs. When the electric field exceeds the turn-on electric field, the escaped electrons increase rapidly.
For the emitting layer (AlN), driving voltage provides an accelerating electric field for the electrons, and almost all the energy from the driving source is used to excite the light-emitting layer. After escaping from the CNTs, the electrons are accelerated by the electric field between the cathode and the anode and gain kinetic energy. Then the electrons penetrate into the AlN and generate electron-hole pairs through impact-ionization processes. The excited electrons relax to a lower state via radiative recombination and result in photon emission. As displayed in Figure 6 in revised manuscript, a higher driving voltage will enable the electrons escaped from CNTs to get higher energy, allowing them to penetrate deeper into the sample and generate more electron-hole pairs, resulting a stronger emission.
- In other words, how long can the device in this work maintain the light emission intensity with the same driving voltages, that is, the lifetime of the device?
For the practical application, it is better to show the CNT-based electron emitted source lifetime and AlN layer one to demonstrate the stability of the electron beam source and the emission material.
Response: It is a pity for us that we have neglected to measure the lifetime of the device, and it is difficult for us to supplement the experiment within the short period of time for paper revision. Although a data for lifetime of the device has not been recorded, we have noticed that the degradation of the emission has not been observed for more than 1 hour when we kept the voltage fixed at 3.92 kV to collect the CL spectra and measure the emission output power.
We believe that the stability of device mainly depends on the current stability of CNTs. The CNTs are the cathode material with mature growth technology in our laboratory. Here, we have measured the DC field emission stability of CNTs lasting for 30 min, as shown in Figure R1. The emission current is stable, whose current fluctuation ratio is 3.9%, which is defined as σ = (Imax – Imin) / (Imax + Imin). The stable emission current from CNTs ensures the stable electron beam excitation and thus stable light emission. The current stability is one of the reasons why we take CNTs as cathode in our device.

Figure R1. The DC stability measured at 1 mA lasting for 30 min.
- This paper explains that the emission at 285 nm (4.35 eV) originates from the transition from the conduction band to the aluminum vacancy and vacancy-impurity complex.
While the emission at 330 nm (3.75 eV) is attributed to the transition between the shallow donor and aluminum vacancy complex.
What is the bandgap in your AlN material?
Response: The bandgap in our AlN material is measured to be 6.11 eV. The Tauc Metod [R2] was used to determine the bandgap of semiconductors, which is based on the absorption coefficient α and can be described by the equation:
(αhν)n = B(hν - Eg)
where hν is photon energy, B is a constant, Eg is the bandgap, n = 2 for AlN material (direct bandgap semiconductor).
As shown in Figure R2, an absorption spectrum of the AlN sample has been obtained, and the inset exhibits corresponding Tauc plot with a linear fit extrapolated to the x-axis. According to the above equation, the x-axis intersection point of the linear fit gives an estimate of the bandgap value of AlN sample, namely 6.11 eV.

Figure R2. The absorption spectrum of the AlN sample. The inset is corresponding Tauc plot.
In the CL process, the transition behavior of electrons can be described as follows. When the AlN emitting layer is excited, most of the excited electrons is captured by shallow donor, and then recombine with the holes on the energy levels related to aluminum vacancy complex, resulting an emission at 330 nm (3.75 eV). A small number of electrons are excited to the conduction band and contribute to the emission at 285 nm (4.35 eV) by recombining with holes on energy levels related to aluminum vacancy complex. No conduction band electron recombines with holes in valence band, so the near band-gap emission of AlN is absent in the CL spectra.
Figure R2 has been added in revised manuscript as Figure 3. The sentences “The Tauc method [33] was used to determine the bandgap of the AlN sample, which is based on the absorption coefficient α and can be described by the following equation:
(αhν)n = B(hν - Eg) (2)
where B is a constant, ν is the frequency of the photon, h is the Plank constant, Eg is the bandgap, n = 1/2 for indirect bandgap semiconductor and n = 2 for direct bandgap semiconductor. As the AlN is a direct bandgap semiconductor, we take n = 2. As shown in Fig. 3, an absorption spectrum of the AlN sample has been obtained, and the inset exhibits the corresponding Tauc plot with a linear fit extrapolated to the x-axis. According to the above equation, the x-axis intersection point of the linear fit gives an estimate of the bandgap value of the AlN sample, which is measured to be 6.11 eV.” have been added at line-165 in revised manuscript.
The sentences “More specifically, the transition behavior of electrons can be described as follows. When the AlN emitting layer is excited, most of the excited electrons are captured by the shallow donor, and then recombine with the holes on the energy levels related to the aluminum vacancy complex, resulting in a strong emission at 330 nm (3.75 eV). A small number of electrons are excited to the conduction band and contribute to the weak emission at 285 nm (4.35 eV) by recombining with holes on energy levels related to the aluminum vacancy complex. No conduction band electron recombines with holes in the valence band, so the near band-gap emission of AlN is absent in the CL spectra.” have been added at line-221 in revised manuscript.
- I wonder whether we could increase the PL intensity of the device in this work by increasing aluminum vacancy concentration or not. Would you answer this?
Response: We think increasing aluminum vacancy concentration can not increase the emission intensity under present excitation voltage. Aluminum vacancy related defects are the low energy levels in the radiative recombination. All the electrons from the conduction band (4.35 eV) and the shallow donor (3.75 eV) combine with the holes from aluminum vacancy complex and generate photons. And the spectra do not exhibit saturation phenomenon (for example, the emission at 330 nm does not increase with the increasing driving voltage, and an emission attributed to the transition from the shallow donor to the valence band begin to appear). It means the number of energy levels in aluminum vacancy complex is more than the holes participating radiative recombination. In other words, the aluminum vacancy concentration is enough under current excitation voltage.
- PL peaks in AlN related to aluminum vacancy and vacancy-impurity complex defects. On the other hand, the previous work indicated the oxygen-defect-assisted PL peaks at 2.53 eV. (J. Appl. Phys. 2002, 41, L28).
Are the PL or EL peaks dependent on the defect types, that is, could we control these PL peaks with defect types?
Response: The PL or EL peaks are dependent on the defect types, and we can control these PL peaks with defect types. For example, Nakarmi et al. [R3] have reported an emission peak at 4.70 eV, attributed to the transition of electrons bound to triply charged nitrogen vacancies to neutral Mg impurities in Mg-doped AlN epilayers. Nepal et al. [R4] have reported an emission peak at 5.40 eV, attributed to the transition of free electrons to neutral Zn acceptors, and an emission peak at 4.50 eV, assigned to a DAP transition of electrons bounded to nitrogen vacancies with three positive charges to neutral Zn acceptors in Zn-doped AlN epilayers.
- With 4 kV driving voltage, the penetration depth is simulated by the Monte Carlo method around 200 nm.
Was the sample quality influenced by the voltage of 4 kV?
If we use the higher voltage, do you expect to see a higher intensity of UV emission?
Response: For solid-state semiconductor, the heat buildup and carbon deposition on the sample are the major causes of material damage when exposed to the electron beam with high current densities or for a long time [R5]. In our work, there is no obvious change in sample quality under our driving voltage.
As shown in Fig 5(c) in revised manuscript, the light output intensity increases with the excitation voltage and dose not exhibit saturation phenomenon. With a higher voltage, which can enable more electrons to escape from the cathode and participate in the CL process with higher energy, a higher intensity of UV emission is expected to see. Notably, the voltage should be controlled within a reasonable range, as excessive electric field and current density may result in AlN damage caused by heat accumulation and carbon deposition.
The sentences “Moreover, the light output intensity in Fig. 5c did not exhibit a saturation phenomenon under the present driving voltage, indicating that the output intensity has a large promotion space. With a higher voltage controlled within a reasonable range, which can enable more electrons to escape from the cathode and participate in the CL process with higher energy, a higher intensity of ultraviolet emission is expected.” have been added at line-280 in revised manuscript.
References
[R1] Sveningsson, M.; Morjan, R.E.; Nerushev, O.A.; Sato, Y.; Bäckström, J.; Campbell, E.E.B.; Rohmund, F. Raman Spectroscopy and Field-Emission Properties of CVD-Grown Carbon-Nanotube Films. Appl Phys A Mater Sci Process 2001, 73, 409–418, doi:10.1007/s003390100923.
[R2] MakuÅ‚a, P.; Pacia, M.; Macyk, W. How To Correctly Determine the Band Gap Energy of Modified Semiconductor Photocatalysts Based on UV-Vis Spectra. Journal of Physical Chemistry Letters 2018, 9, 6814–6817.
[R3] Nakarmi, M.L.; Nepal, N.; Ugolini, C.; Altahtamouni, T.M.; Lin, J.Y.; Jiang, H.X. Correlation between Optical and Electrical Properties of Mg-Doped AlN Epilayers. Appl Phys Lett 2006, 89, doi:10.1063/1.2362582.
[R4] Nepal, N.; Nakarmi, M.L.; Jang, H.U.; Lin, J.Y.; Jiang, H.X. Growth and Photoluminescence Studies of Zn-Doped AlN Epilayers. Appl Phys Lett 2006, 89, doi:10.1063/1.2387869.
[R5] Cuesta, S.; Harikumar, A.; Monroy, E. Electron Beam Pumped Light Emitting Devices. J Phys D Appl Phys 2022, 55.

Reviewer 2 Report
The manuscript describes the principle, realization, operation and performance of an ultraviolet light source based on electron-beam driven cathodoluminescence. It focuses on the carbon nanotube electron emitter and aluminum nitride photoemitting anode. The use of AlN wide band-gap semiconductor material is what goes beyond already reported similar lamps (e.g., hBN-based devices).
The paper is well written and provides profound understanding of the device. There are only a few remarks which should be considered:
1. The results and discussion section is rather large, and might better be structured into subsections, e.g., regarding CNT emitter, AlN film cathode, spectral emission analysis.
2. Figure 4 (b) is rather qualitative. It raises more questions instead of answering some, i.e., what is the explanation for the visible crescent moon inhomogeneity, why does it change (rotate) with voltage? The quantitative emission is fully characterized in Figure 4(c), so 4(b) does not add information to the reader.
3. The authors should correct where references are erroneously positioned in the text after the full stop (e.g. lines 48, 53, 65, 112, and others).
4. All emission intensities in the article are given in a.u., however, in the supplement material an absolute estimation for the reading of the power meter is mentioned. At least some estimation of the total radiative emission power of the device might be useful to be added to the article.
5. Can the authors comment on the stability of device? Was there any degradation of the emission observed?
Author Response
Dear reviewer,
Thank you for the enlightening suggestions and recommendations. The following replies have thoroughly addressed all of the comments in the revised manuscript.
Comments and Suggestions for Authors
The manuscript describes the principle, realization, operation and performance of an ultraviolet light source based on electron-beam driven cathodoluminescence. It focuses on the carbon nanotube electron emitter and aluminum nitride photoemitting anode. The use of AlN wide band-gap semiconductor material is what goes beyond already reported similar lamps (e.g., hBN-based devices).
The paper is well written and provides profound understanding of the device. There are only a few remarks which should be considered:
- The results and discussion section is rather large, and might better be structured into subsections, e.g., regarding CNT emitter, AlN film cathode, spectral emission analysis.
Response: Thanks for your suggestion. The results and discussion section has been structured into four subsections: 3.1 CNT emitter, 3.2 AlN film, 3.3 Prototype device, 3.4 Spectra analysis.
- Figure 4 (b) is rather qualitative. It raises more questions instead of answering some, i.e., what is the explanation for the visible crescent moon inhomogeneity, why does it change (rotate) with voltage? The quantitative emission is fully characterized in Figure 4(c), so 4(b) does not add information to the reader.
Response: The photos in Figure 4(b) were taken with a hand-held camera at the exit window of the vacuum chamber when the device was in operation. The crescent moon inhomogeneity was caused by the ultraviolet lens used to improve light beam divergence (shown in Figure 3) when our camera was placed off the imaging plane. The rotation of light spot was caused by the slight rotation of the hand-held camera.
Although the quantitative emission is fully characterized in Figure 4(c), we believe that a photo of the light emission is necessary which can enable the readers to gain an intuitive understanding of the device luminous effect. The photo has been carefully retaken using an ultraviolet sensitive camera, as shown in Figure R1(b). In the photo, the outer circle was caused by light scattering from the edge of the ultraviolet lens. The luminescent spot of the device was located inside the circle, with a shape of cross which was caused by four symmetrically distributed CNT cubes (Figure 4(b) in revised manuscript).
The original Figure 4 (Figure 5 in revised manuscript) has been replaced with Figure R1. The sentences “A photo of the light emission is necessary which can enable the readers to gain an intuitive understanding of the device luminous effect. As shown in Fig. 5b, a photo of the device was taken using an ultraviolet-sensitive camera when the device was in operation. In this photo, the outer circle was caused by the light scattering from the edge of the ultraviolet lens used to improve light beam divergence (shown in Fig. 4a). The luminescent spot of the device was located inside the circle, with a shape of cross which was caused by four symmetrically distributed CNT cubes (shown in Fig. 4b).” have been added at line-229 in revised manuscript to replace the original sentences “Additionally, the light output intensity increases with the excitation voltage. As shown in luminous photographs (Fig. 4b), when the device began to give out light under driving voltage V1, only a slight light can be seen by naked eyes. As the voltage increases (V2 and V3), the device exhibits brighter output light, which is consistent with the content expressed in Fig. 4c.”

Figure R1
- The authors should correct where references are erroneously positioned in the text after the full stop (e.g. lines 48, 53, 65, 112, and others).
Response: Thanks for your suggestion. We have corrected these mistakes in revised manuscript.
- All emission intensities in the article are given in a.u., however, in the supplement material an absolute estimation for the reading of the power meter is mentioned. At least some estimation of the total radiative emission power of the device might be useful to be added to the article.
Response: Thanks for your suggestion. The sentences “To estimate the power efficiency (PE) of the ultraviolet device, the light output power is required to be measured. A schematic diagram of the light output power measurement setup is shown in Fig. S3. The light emitted from AlN thin film first passed through a 1-inch ultraviolet lens to improve beam divergence, and then exited from the vacuum chamber through a 2-inch sapphire window. A 2-inch ultraviolet lens was placed close to the sapphire window to focus the light beam. Finally, the output power was measured by a power meter. When a driving voltage with an amplitude of 3.92 kV and a duty ratio of 10% (namely, Vd = 3.92 kV and δ = 10%) was applied, the corresponding current was 1.6 mA (Id = 1.6 mA) with a same duty ratio. Thus, the input electrical power (Pin) was about 62 mW, which can be calculated using the expression Pin = Id Vd δ2. Under this driving voltage, the light output power (Pout) was measured to be about 10 μW. Consequently, the power efficiency (PE) of the device, calculated by the equation PE = Pin / Pout was estimated to be about 0.02%.” has been added at line-235 in revised manuscript.
- Can the authors comment on the stability of device? Was there any degradation of the emission observed?
Response: The stability of device mainly depends on the current stability of CNTs. As the CNTs are the cathode material with mature growth technology in our laboratory, previous works have proved its current stability [R1], which is one of the reasons why we take CNTs as cathode. Here, we have measured the DC field emission stability of CNTs lasting for 30 min, as shown in Figure R2. The emission current is stable, whose current fluctuation ratio is 3.9%, which is defined as σ = (Imax – Imin) / (Imax + Imin).
In fact, the degradation of the emission has not been observed for more than 1 hour when we kept the voltage fixed at 3.92 kV to recode the CL spectra and measure the emission output power.

Figure R2. The DC stability measured at 1 mA lasting for 30 min.
References
[R1] Zhu, W.; Zhang, Y.; Xu, N.; Tan, Y.; Zhan, R.; Shen, Y.; Xu, Z.; Bai, X.; Chen, J.; She, J.; et al. Epitaxial Growth of Multiwall Carbon Nanotube from Stainless Steel Substrate and Effect on Electrical Conduction and Field Emission. Nanotechnology 2017, 28, 305704, doi:10.1088/1361-6528/AA780C.

Round 2
Reviewer 1 Report
The authors have fully addressed the mentioned comments in the revised manuscript. This version is suitable for publication in Nanomaterials.